# Molecular Design, Spectroscopic, DFT, Pharmacological, and Molecular Docking Studies of Novel Ruthenium(III)–Schiff Base Complex: An Inhibitor of Progression in HepG2 Cells

**DOI:** 10.3390/ijerph192013624

**Published:** 2022-10-20

**Authors:** Amani F. H. Noureldeen, Safa W. Aziz, Samia A. Shouman, Magdy M. Mohamed, Yasmin M. Attia, Ramadan M. Ramadan, Mostafa M. Elhady

**Affiliations:** 1Biochemistry Department, Faculty of Science, Ain Shams University, Cairo 11566, Egypt; 2Department of Laboratory and Clinical Sciences, College of Pharmacy, University of Babylon, Babylon 51002, Iraq; 3Cancer Biology Department, National Cancer Institute, Cairo University, Cairo 12613, Egypt; 4Chemistry Department, Faculty of Science, Ain Shams University, Cairo 11566, Egypt

**Keywords:** ruthenium–Schiff base complex, DFT studies, antitumor activity, DNA damage, cell cycle arrest, apoptosis, molecular docking

## Abstract

A novel ruthenium(III)–pyrimidine Schiff base was synthesized and characterized using different analytical and spectroscopic techniques. Molecular geometries of the ligand and ruthenium complex were investigated using the DFT-B3LYP level of theory. The quantum global reactivity descriptors were also calculated. Various biological and molecular docking studies of the complex are reported to explore its potential application as a therapeutic drug. Cytotoxicity of the complex was screened against cancer colorectal (HCT116), breast (MCF-7 and T47D), and hepatocellular (HepG2) cell lines as well as a human normal cell line (HSF). The complex effectively inhibited the tested cancer cells with variable degree with higher activity towards HepG2 (IC_50_ values were 29 μM for HepG2, 38.5 μM for T47D, 39.7 μM for HCT, and 46.7 μM for MCF-7 cells). The complex induced apoptosis and cell cycle arrest in the S phase of HepG2 cells. The complex significantly induced the expression of H2AX and caspase 3 and caspase 7 gene and the protein level of caspase 3, as well as inhibited VEGF-A and mTOR/AKT, SND1, and NF-kB gene expression. The molecular docking studies supported the increased total apoptosis of treated HepG2 cells due to strong interaction of the complex with DNA. Additionally, the possible binding interaction of the complex with caspase 3 could be responsible for the elevated activity of caspase 3–treated cells. The score values for the two receptors were −3.25 and −3.91 kcal/mol.

## 1. Introduction

Globally, hepatocellular carcinoma (HCC) is one of the leading lethal tumors worldwide; it is the third cause of cancer-related death [1]. Its incidence, morbidity, and mortality are high, especially in Asia and Africa [2]. HCC is multifactorial in etiology, where contamination of foodstuff to hepatic carcinogens such as aflatoxin occurs. In addition, the endemic high prevalence of hepatitis B and hepatitis C strongly predisposes to the development of chronic liver disease and the subsequent development of HCC [3]. Surgery and chemotherapy are the current standard curative methods for patients with HCC at an early stage, but the prognosis is still flawed. Therefore, other effective therapeutic strategies with lower costs and fewer side effects are urgently required for the treatment of HCC.

Platinum-based drugs, such as cisplatin, oxaliplatin, and carboplatin, are widely used in chemotherapy treatments. They play an important role in fighting many types of cancer, such as testicular, ovarian, bladder, cervical, hepatic, and lung cancers. They obstruct the nucleic acid (DNA) from replicating and from acting as a route for protein making. However, these platinum-based drugs cause terrible side effects, including severe nausea, vomiting, critical anemia, hair loss, and susceptibility to infections, and they destroy the patients’ kidneys and livers. Thus, the problems of such severe symptoms have encouraged the development of novel metal complexes that are able to interact with DNA and have potential anticancer activity along with light side effects. Therefore, researchers seek out other metal-based drugs, such as gold(III), gold(I), ruthenium(II), and ruthenium(III) complexes. In particular, ruthenium complexes have received a lot of attention as an alternative to platinum-based drugs due to their good biodistribution and multimodal actions [4,5,6]. Numerous ruthenium complexes have exhibited remarkable antitumor activity, and some of them have various advantages over platinum drugs, such as potent efficacy, less drug resistance, and low toxicity [7,8,9]. Ruthenium complexes have proved to be successfully able to penetrate tumor cells and to bind effectively to nucleic acids. Thus, ruthenium complexes present great research subjects, and they are tested for their effects versus different cell lines of cancer. These complexes are noted to be promising substitute selects instead of cisplatin and its derivatives, and they are expected to become a new generation of clinical metal antitumor drugs. Ruthenium derivatives are generally less toxic than other platinum drugs, and they are most efficient against drug resistance induced by other drugs. Particularly, ruthenium complexes of Schiff base ligands have succeeded to show good achievement for biological applications as antioxidant reagents and antimicrobial and anticancer potential medications. This is probably because of the octahedral coordination preference of both ruthenium(II) and ruthenium(III) complexes. Such three-dimensional geometrical frameworks may provide potential elevation of site selectivity for better binding to the biologically active macromolecular receptors. These molecular structure arrangements may also cause deactivation of a tumor suppressor gene (p53) in the cells of cancer. Furthermore, they can overcome bad chemotherapeutic results and clinical drug resistance [10,11,12]. On the other hand, the extraordinary role of the pyrimidine ring in dihydropyrimidine is presumed to be an essential part in nucleic bases, vitamins, enzymes, chlorophyll, hemoglobin, and hormones. In addition, pyrimidine derivatives in medicinal chemistry have attracted much attention due to their availability in the substructures of therapeutic natural products. They have been identified for their therapeutic applications, and several pyrimidine derivatives have been developed as chemotherapeutic agents and found to have wide clinical applications, such as anti-inflammatory, antibacterial, and anticancer agents [13,14]. Previous works have shown that the combination of ruthenium and pyrimidine derivatives results in interesting bioactive species. For example, many reports have shown that complexes of ruthenium with pyrimidine derivatives, such as 6-amino-5-(aryldiazenyl)-*N*1,*N*3-dimethyl-2-thioxo-pyrimidin-4-one [15] and triazolopyrimidine derivatives [16], have promising applications as anti-HIV and anticancer reagents through the interaction with the DNA.

The development of new antitumor drugs requires understanding the mechanisms of action of new pharmacological candidates, which highlights the importance of elucidating the molecular and biochemical mechanisms involved. For example, some ruthenium complexes target the genomic DNA and may cause cell cycle arrest, ROS-mediated mitochondrial alterations, and cell death by apoptosis [17]. It is worth mentioning that there are already some ruthenium complexes that have entered clinical trials. The first candidate was an imidazolium complex [RuCl_4_(imidazole)(DMSO)] (DMSO (dimethyl sulfoxide)), which reached the phase II stage study on human beings [18]. Another promising Ru(III) complex that have entered clinical trials is an indazole complex [trans-RuCl_4_(indazole)_2_] [19,20].

Metal-based therapeutics containing Schiff base moieties could offer versatile electronic and structural features, including a range of oxidation states, coordination geometries, type, and number of other ligands. Thus, metallodrugs undergo activation by ligand substitution or redox reactions, and they are multitargeting, which need to be considered to establish structure–activity relationships. One of the most common diseases that have deeply increased research is cancer. The use of virtual techniques, such as molecular docking, is now common in evaluating the suitability of new drugs. In molecular docking, analysis of the docking data is useful in predicting the conformational changes associated with the amino acid residues at the binding positions to accommodate the docked hydrophobic inhibitors. A large number of articles and reviews have reported the molecular docking of transition metal complexes with many macromolecular bioactive targets. Among those articles, molecular docking of transition metal complexes of Schiff bases has widely appeared [19,20,21,22]. Many of these metal–Schiff base derivatives have promising applications as potential drugs for the treatment of antimicrobial, antiviral, and anticancer diseases.

Our interest in synthesizing many Ru(II)– and Ru(III)–Schiff base complexes and investigating their biological and medicinal activity as potential anticancer agents [6,11,21,22] has prompted us to design more novel Schiff base–ruthenium complexes. The unique characteristics of ruthenium–pyrimidine base derivatives as promising bioactive agents encouraged us to synthesize and investigate a novel Ru–pyrimidine Schiff base. Here, we report the synthesis, spectroscopic, DFT, pharmacological, and molecular docking studies of a ruthenium(III)–pyrimidine Schiff base complex (**RuL**, **Ru** = RuCl_3_(H_2_O); **L** = Schiff base ligand, Figure 1) as it was found to be an inhibitor of progression in HepG2 cells. This complex has proved its novelty as a potential anticancer agent that it showed excellent in vitro cytotoxic activity on several cell lines, especially those of HepG2 cells.

## 2. Experimental

### 2.1. Materials and Instruments

Ruthenium(III) chloride, 2-amino-4,6-dimethylpyrimidine, and 2-chloro-5-nitrobenzaldehyde were purchased from Aldrich. All the solvents were of analytical reagent grades. IR measurements (KBr pellets) were carried out on a Unicam-Mattson 1000 FTIR spectrometer (Pye-Unicam, Cambridge, UK). NMR measurements were performed on a Spectrospin-Bruker 300 MHz spectrometer. Samples were dissolved in deuterated dimethyl sulfoxide, (CD_3_)_2_SO, with the use of tetramethylsilane, TMS, as an internal reference. Conductivity measurements (0.5–1 × 10^−3^ M solutions in DMF at 25 °C) were measured using a Jenway 4010 conductivity meter. Elemental analyses were performed on a PerkinElmer 2400 CHN elemental analyzer. Mass spectrometry measurements of the solid compounds (70 eV, EI) were carried out on a Finnigan MAT SSQ 7000 spectrometer.

### 2.2. Synthesis of 2-Chloro-5-Nitrophenyl-(4,6-Dimethylpyrimidinyl)methanimine Schiff Base; L

A mixture of ethanolic solution (30 mL) of 2-amino-4,6-dimethylpyrimidine (1.23 g, 10 mmol) and 2-chloro-5-nitrobenzaldehyde (1.85 g, 10 mmol) was refluxed for 3 h. The resulting pale-yellow solution was left to stand overnight at room temperature. The isolated yellowish-white solid product was filtered off and washed several times with ethanol. The residue was then recrystallized from hot ethanol to give fine crystals of the compound (yield 92%).

C_13_H_11_N_4_O_2_Cl. Elemental analysis, found (Calc.): % C, 53.63 (53.71); % H, 3.85 (3.81); % N, 19.14 (19.27). Mass spectrometry: M.M = 290.71; m/z = 291, 292. IR data, cm^−1^: νC=N, 1596(s), 1577(s); ν_as_NO_2_, 1523(s,b); ν_s_NO_2_, 1346(s). NMR data, ppm: 10.32 s (1H, CH=N), 8.53–6.31 m (4H, -Ph), 2.23 s (3H, -CH_3_), 2.16 d (3H, -CH_3_).

### 2.3. Synthesis of the (trichloro)(monoaquo)(2-Chloro-5-Nitrophenyl-(4,6-Dimethylpyrimidinyl)-Methanimine) ruthenium(III) Complex; RuL

One millimole solution of metal salt (RuCl_3_.3H_2_O, 0.26 g) in 20 mL EtOH was added to 1 mmol solution of the Schiff base (0.29 g) dropwise, and then refluxed with stirring for 5 h at 80 °C. The obtained precipitate was filtered off and washed with ethanol. The crude was recrystallized from hot ethanol, and fine brown crystals were obtained. The crystals were left to dry in air for a few hours (yield 78%).

C_13_H_13_N_4_O_3_Cl_4_Ru. Elemental analysis, found (Calc.): % C, 30.35 (30.25); % H, 2.66 (2.54); % N, 10.65 (10.86); % Cl, 27.30 (27.47). Mass spectrometry: M.M = 516.14; m/z = 516, 517, 518, 519. IR data, cm^−1^: νOH, 3417(s,b); νC=N, 1719(s), 1639(s), 1602(s); ν_as_NO_2_, 1523(m); ν_s_NO_2_, 1346(m); νRu-N, 574(w), 519(w).

### 2.4. Computational Details

All calculations were performed using the hybrid density functional theory (DFT/B3LYP) method as implemented in the Gaussian09 software package [23]. The geometry of the ligand atoms was optimized using the standard double zeta plus polarization basis set 6–31G (d,p), while the effective core potential basis set LANL2DZ was used for the ruthenium complex. The purpose of the quantum mechanics calculations is to validate the proposed three-dimensional (3D) structure of the ligand and Ru complex and to find key factors for their activities.

### 2.5. In Vitro Studies

#### 2.5.1. Cancer Cell Lines

To test the anticancer activity of the reported ruthenium complex (**RuL**), four human cancer cell lines (breast, MCF-7 and T47D; colorectal, HCT116; and hepatocellular, HepG2, along with normal human splenic fibroblasts, HSF) were screened. They were originally obtained from the American Type Culture Collection (ATCC; Washington, DC, USA), and then were maintained by serial subculturing at the National Cancer Institute (NCI, Cairo, Egypt).

#### 2.5.2. Cytotoxicity Assay

The cytotoxicity of the reported **RuL** was examined using cancer MCF-7, T47D, HCT116, and HepG2 cells lines as well as normal cell HS. The cancer cell lines were maintained in RPMI-1640 supplemented with 10% fetal bovine serum and 2 mM L-glutamine, 100 µg/mL streptomycin, and 100 U/mL penicillin and incubated in a humidified 5% CO_2_ atmosphere. Cell viability was assayed using sulforhodamine B (Sigma-Aldrich, St. Louis, MO, USA) method, as previously described [24]. In brief, 1.5 × 10^3^ cells/well were seeded in 96-well plates; after 24 h, cells were exposed to different concentrations (10–100 µM) of the compound for 48 h. Absorbance at 540 nm was measured with an ELISA reader (Tecan Sunrise, Männedorf, Switzerland). Each concentration was repeated in triplicate, and the experiment was repeated three times. Cell viability was expressed as the percentage of absorbance of the compound-treated cells relative to that of the vehicle-treated cells. The IC_50_ and 95% CL were evaluated. According to the results, the **RuL** showed the largest cytotoxicity effect against HepG2 cells.

#### 2.5.3. Annexin V/Propidium Iodide Staining for Apoptosis Assessment

The following tests were carried out using HepG2 cells and the **RuL** complex. An Annexin-V-Fluos staining kit (Fisher Scientific, Hampton, NH, USA) was used according to the manufacturer’s instruction to test the effect of the complex on apoptosis. The kit has double stains where apoptotic cells and necrotic cells were stained with annexin V (green fluorescence) and propidium iodide (PI, red fluorescence), respectively. HepG2 cells were grown to ~70% confluence and treated with 29 µM of the **RuL** (equivalent to IC_50_ value of the complex) for 48 h. After incubation, the floating and adherent cells were collected and washed three times with cold phosphate-buffered saline (PBS), followed by the addition of 5 μL of annexin V and PI. The cells were incubated for 20 min and then analyzed by FACScan, Beckman Coulter Epics XL Flow Cytometer (Beckman, Brea, USA).

#### 2.5.4. Cell Cycle Analysis

HepG2 cells were seeded for 24 h at 70% confluence at 37 °C and 5% CO_2_. Cells were treated for 48 h with a concentration of IC_50_ of the **RuL** complex (29 µM). After the treatment incubation, the floating cells were aspirated and discarded. The cells were detached using trypsin, then washed two times with cold PBS, followed by centrifugation. The pellet was resuspended with propidium iodide for 20 min for flow cytometry analysis. Flow cytometry was performed with a FACScan flow cytometer (Beckman Coulter Epics XL Flow Cytometer). A minimum of 10,000 cells/sample were collected, and cell cycle analysis was conducted.

#### 2.5.5. Expression Levels of Caspase 3, VEGF-A, mTOR, NF-kB, and SND1 by RT-PCR

HepG2 cells were seeded in 6-well plates and, after 24 h cells, were treated with IC_50_ concentration of **RuL** for 48 h. After the incubation time, cells were trypsinized and collected in tubes and then centrifuged for 5 min at 1200 rpm. The cellular total RNA was isolated from cells with a TRIzol reagent (Invitrogen, Carlsbad, CA, USA), and cDNA was obtained from 1 μg of total RNA using a SuperScript II reverse transcription kit, according to the manufacturer’s instructions (Invitrogen, Carlsbad, CA, USA). The cDNA was then amplified by PCR on a CFX384 Touch Real-Time PCR Detection System (Bio-Rad Laboratories, Hercules, CA, USA) using SYBR Green PCR Master Mix (Promega, Fitchburg, WI, USA), according to the manufacturer’s protocol. Fast amplification parameters were as follows: one cycle of 95 °C for 10 min, followed by 40 cycles of 95 °C for 15 s and 60 °C for 1 min. The primer sequences are shown in Appendix A. Quantitative analysis of data was performed as described previously. Values were normalized to GAPDH and are shown as relative expression levels.

#### 2.5.6. Assay of Protein Levels of Caspase 3, VEGF-A, mTOR, NF-kB, and SND1

The protein levels of caspase 3, VEGF-A, mTOR, NF-kB, and SND1 were assessed spectrophotometrically at 450 nm in cell lysate using ELISA kit (Invitrogen, Carlsbad, CA, USA) following the manufacturer’s instructions. Cells were cultured in 75 cm^3^ flasks and left till 70–80% confluent; cells were treated with the **RuL** complex for 48 h. Then, the treated and control cells were lysed in a RIPA lysis buffer containing protease inhibitors. Each concentration was repeated two times, and the experiment was carried out three independent times. The activity was calculated relative to the corresponding protein content.

### 2.6. Molecular Docking Studies

Molecular docking studies were carried out using the Molecular Operating Environment (MOE) software package, version 2014.09. The macromolecule targets B-DNA (PDB ID: 1BNA) and caspase 3 (PDB ID: 3KJF) were used for docking the ruthenium complex. 1BNA is an X-ray crystal structure of a B-DNA (dodecamer d (CGCGAATTCGCG)_2_) running a 3′–5′ direction [25]. 3KJF is a caspase 3 protein, which has a central role in programmed cell death [26]. Structures of the targets were energetically optimized after inserting hydrogen atoms. The resulting model afforded systematic conformational research with an RMS gradient of 0.01 kcal/mol.

### 2.7. Statistical Analysis

All data are expressed as mean ± SD unless otherwise specified. Differences between treated samples and untreated controls were analyzed by *t*-test. Statistical analyses were performed using GraphPad Stat, version 7.03 (GraphPad, San Diego, CA, USA). Statistical significance was set at *p* < 0.05.

## 3. Result and Discussion

The reported Schiff base ligand and its ruthenium complex were synthesized and characterized using different spectroscopic techniques (IR, ^1^H NMR, mass) and elemental analyses. Although the ligand and its ruthenium complex were isolated as fine crystals, attempts to isolate crystals suitable for single crystal X-ray analysis were unsuccessful. The mass spectra of the two compounds showed the parent molecular peaks signals with the expected pattern (Appendix A). The molar conductivity values, Λm, of 0.5–1 × 10^−3^ M solutions of the complex at 25 °C were found to be in the range of 15–18 ohm^−1^mol^−1^cm^2^, indicating that the complex is nonelectrolyte. The mass spectrum of the Schiff base ligand displayed the parent molecular ion peak, (P)^+^ = 291, and a peak at (P + 1)^+^ = 292. On the other hand, the mass spectrum of RuL complex showed a pattern of signals due to the most abundant isotopes of ruthenium ion (^100^Ru, ^101^Ru, and ^102^Ru). The peaks appeared at m/z = 516, 517, 518, and 519 corresponding to the molecular ion peaks (P-1)^+^, (P)^+^, (P + 1)^+^, and (P + 2)^+^ ions, respectively. The presence of a water molecule in the complex was indicated from its mass spectrum as it gave a signal at m/z = 500 due to the molecular ion (P-H_2_O)^+^. The obtained elemental analyses and spectroscopic data (Experimental section) of the two compounds were in accordance with the proposed molecular formulas.

### 3.1. Spectroscopic Studies

The IR spectra (KBr pellets) of the reported compounds were recorded in the region 4000–400 cm^−1^ (Appendix A). The most prominent IR bands of the important functional groups, such as νOH, νC=N, ν_as_NO_2_, and ν_as_NO_2_, are given in the Experimental section. The IR spectrum of the ligand displayed bands at 1596 and 1577 cm^−1^ due to stretching frequencies of two distinct C=N bonds, Figure 1. Interestingly, the IR spectrum of the Ru complex exhibited three bands due to three different C=N bonds (Figure 1); these bands were shifted to higher wavenumbers as expected. It is obvious that the two spectra are almost similar in the region 1700–1400 cm^−1^ with the appropriate shift due to complex formation, while they are different in the lower wavenumbers as fingerprint to every compound. In addition, the IR spectrum of the complex displayed nonligand bands corresponding to Ru–N bonds [27]. The ^1^H NMR spectrum of the ligand (Appendix A) displayed a sharp singlet at 10.32 ppm due to the proton of the azomethine moiety [27,28]. In addition, the spectrum exhibited multiplets due to the phenyl protons and signals due to the methyl groups.

### 3.2. Stereochemistry and Chemical Reactivity Prediction

The optimized structural, geometrical parameters and energetics of the ground state for the Schiff base ligand and its ruthenium complex were calculated using density functional theory (DFT). The structure of the ligand was optimized using the B3LYP/6-311G (double zeta) level of theory, while the effective core potential basis set LANL2DZ was used for the optimization of the ruthenium complex. In order to investigate the stereochemistry of the most stable structures, it was focused first on the structure of the Schiff base molecule, more specifically, on the orientation of its functional groups with respect to each other and with respect to the central azomethine (HC=N) moiety. Table 1 gives the important bond lengths and bond angles for both the ligand and ruthenium complex. All the calculated bond lengths and bond angles were in the normal range observed before [29,30,31]. The energetically stable model for the ligand with a minimization energy of 23.96 kcal/mol showed specific features, where the molecule is non-planar. The dihedral angles N(13)-C(12)-N(15)-C(8), C(4)-C(5)-C(8)-N(15), and C(5)-C(8)-N(15)-C(12) were 73.1°, 137.9°, and 176.1°, respectively (Figure 1A). The two bond distances involved in the azomethine group, C(12)-N(15) and C(8)-N(15), were 1.40 and 1.28 Ǻ, respectively. Obviously, the latter bond is shorter due to the double bond characteristic. The bond angles of the azomethine part, C(5)-C(8)-N(15), N(11)-C(12)-N(15), and C(8)-N(15)-C(12), were 127.6°, 123.2°, and 131.0°, respectively. The orientation of the functional groups of the ligand suggested that it may only coordinate to the ruthenium ion through the nitrogen of the azomethine and one of the nitrogen atoms of the pyrimidine moiety; that is, it can act as a bidentate ligand with the coordination of the NN set of donors. Figure 1A shows the charge distribution on the different atoms of the ligand. The charge distribution showed that the two nitrogen atoms, N(15) and N(11), have the highly negative charge density relative to other atoms in the molecule (−0.32 and −0.40, respectively). Note that the oxygen atoms of the nitro group are not considered for coordination.

The optimized geometry (minimization energy of 127.91 kcal/mol) and numbering of atoms of the ruthenium complex are presented in Figure 1B. The ruthenium coordinated with the ligand from two nitrogen atom donors (azomethine nitrogen and one of the nitrogen atoms of pyrimidine ring) to form a four-membered chelate. In addition, the Ru completed its six-coordination environment by three chloro and a water molecule to adopt a distorted octahedral structure. The bond length of C(15)-N(11), which was involved in the coordination, was increased relative to that of the ligand itself due to complex formation. Additionally, the bond angle N(11)-C(15)-N(20), which is participated in the chelate, was significantly decreased from 123.2° in the free ligand to 103.5° due to complex formation. Notably, the water molecule coordinated trans to one of the chloro ligands with an angle close to linearity (171.6°). Furthermore, the three chloro ligands coordinated cis to each, and their angles deviated from 90° (around 100°) due to their electron pair repulsion. On the other hand, the bond angle N(11)-Ru(30)-N(20) in the chelate was found to be 64.2°. Although the four-membered chelate is an unfavorable coordination due to the angles strain, the larger size of ruthenium helped in compensating such strain.

The global chemical reactivity parameters, namely, HOMO, LUMO, energy gap (Δ*E*), electronegativity (*X*), chemical potential (*V*), electron affinity (*A*), ionization potential (*I*), chemical hardness (*η*), chemical softness (*S*), and electrophilicity index (*ω*), of the reported compounds are given in Table 2 [32]. The frontier molecular orbital (HOMO and LUMO) energies were estimated using the DFT method (Figure 2). The HOMO orbital energy represents the electron donating ability, while the LUMO orbital energy characterizes the electron withdrawing ability. The energy gap between HOMO and LUMO shows the molecular chemical stability (reactivity); it is a critical parameter for determining molecular electrical transport properties. A smaller energy gap reflects the easiness of the charge transfer (CT) and the polarization, which occurs within the molecule [33]. Accordingly, the complex is more reactive than the ligand as it has a smaller energy gap. Furthermore, the electronegativity parameter is a reflection for the electrostatic potential, where the electron partially transferred from one of lower electronegativity to another of higher electronegativity. The results showed that the ligand has lower *X* than the complex. The chemical potential of a species is the energy that can be absorbed or released due to a change of the particle number of the given species. Its value is arbitrarily equal to the negative value of electronegativity. On the other hand, the results of small chemical hardness values for the two derivatives reflect the ability of charge transfer within the molecule. The order of increasing the charge transfer within the molecule is: **RuL** < **L**. Ionization potential is the minimum energy required to remove the most loosely bound electron from a neutral atom or molecule, while electron affinity is the amount of energy released when an electron is attached to a neutral atom or molecule to form a negative ion. From the HOMO and LUMO energies, ionization potential and electron affinity are expressed as I~-E_HOMO_ and A~-E_LUMO_. The electrophilicity index is a measure of the electrophilic power of a molecule. From Table 2, the ligand has lower electrophilicity power relative to the complex. From the reactivity descriptors shown in Table 2, one can correlate between the structure and the activity of the pyrimidine–Schiff base ligand and its ruthenium complex. The central assumption of the structure–activity relationship (SAR) is that the activity of molecules is reflected in their structure; that is, similar molecules have similar activities. Thus, the SAR concept assumes that the structural properties of a molecule, such as its geometrical and electronic properties, contain the features responsible for its physical, chemical, and biological properties. Therefore, SAR can be used to predict the biological activity of a molecule from its molecular structure. This concept is commonly used in drug discovery to guide the development of desired new compounds. As we can see from Table 2, the reported ruthenium–pyrimidine complex has lower chemical softness relative to that of the corresponding Schiff base ligand, while it affords higher values for the chemical potential and electrophilicity index. The sequence of these descriptors reflects that the complex could have better bioactivity. Therefore, the coordination of Ru(III) species to the pyrimidine–Schiff base has enhanced the biological activity of the pyrimidine base moiety [34].

### 3.3. In Vitro Studies

#### 3.3.1. Cytotoxicity Screening on the Tested Human Cancer Cell Lines

The cytotoxic effect of the **RuL** complex was screened on four different human cancer cell lines (breast, MCF-7 and T47D; colorectal, HCT116; and liver, HepG2) using the sulforhodamine B colorimetric (SRB) assay. The complex manifested antitumor activity on the tested cancer cell lines, as shown in Figure 3A. For comparison, the effect of cisplatin toward the proliferation of HepG2 is also shown in Figure 3B. The results revealed that the **RuL** complex has a broad-spectrum antitumor activity. It could effectively inhibit all the tested human cancer cell lines and inhibit the growth with high anticancer activity. The obtained IC_50_ values were 29 ± 3 μM for HepG2, 38.5 ± 2 μM for T47D, 39.7 ± 5.6 μM for HCT, and 46.7 ± 7 μM for MCF-7 cells. Thus, the **RuL** complex exhibited promising activity on the tested cell lines especially for the HepG2 cell line, as it has smaller IC_50_, which is comparable to that of cisplatin (29 μM for **RuL** versus 22 μM for cisplatin). Notably, the cytotoxic effect of the **RuL** complex on the normal human splenic fibroblasts (HSF) was low as it has an IC_50_ higher than 100 μM (Figure 3A). Furthermore, for the HepG2 cancer cell line, **RuL** exhibited a selectivity index equal to 3.45, which is like that found for the positive control cisplatin. (The selectivity index (SI) is calculated from the formula: SI = IC_50_ of normal cells/IC_50_ of cancer cells.) Based on the SI value, the **RuL** complex showed low toxicity towards the HSF human normal cells and indicates a high selectivity between cancer and normal cells.

#### 3.3.2. Annexin V/Propidium Iodide Staining for Apoptosis Assessment

Cell cycle and apoptosis are regulatory mechanisms of cell growth, differentiation, and development [35]. Mounting evidence shows that cell cycle arrest and apoptosis can reduce cancer cells viability and sensitize cancer cells to chemotherapy or radiotherapy [36,37]. The possible arrest of the cell cycle after exposure of HepG2 cells to IC_50_ dose of the **RuL** complex was analyzed using flow cytometry. As shown in Figure 4, the ruthenium complex caused a reduction in the G0/G1 (98% vs. 88%) phase, accompanied by a corresponding increase in the percentage of cells in the S phase (0.85% vs. 12%). These data suggest that the antiproliferative mechanism for the complex is based on S-phase arrest. Similar results were obtained previously by other ruthenium complexes [38]. One key strategy for the therapy of cancer is to induce cancer cell death through the development of drugs that induce apoptosis [39]. Moreover, the apoptotic effect of **RuL** on HepG2 cells was measured by annexin V-FITC/PI using flow cytometry. Apoptosis and necrosis are two major types of cell death, and their deregulations can result in several diseases, including cancer. The total apoptosis was defined as the sum of early and late apoptosis percentages. Flow cytometry results revealed that the total apoptosis of HepG2 cells treated with the complex was 5.8% versus 0.18% in the untreated control HepG2. The percentage of necrotic cells after treatment with **RuL** was 48% versus 3.3% in the untreated control HepG2, Figure 5.

#### 3.3.3. Expression Levels of Caspase 3 and 7

It is well known that the caspase family proteins function as important regulators in the induction of apoptosis. Caspase 3 and 7 are the key executioner enzymes that are necessary for apoptosis and for normal mammalian life. Caspase 3 is a relatively small protein that consists of two subunits, which contains three and five thiol function groups [40]. Activation of caspase 3 is dependent on its dimerization to a heterotetramer, where the histidine-activated Cys-285 in the active site of the p17 subunit is conserved in the caspase superfamily and is required for enzymatic activity. Caspase 3 is also activated by interaction with metal complexes [38,41]. Based on the obtained data of flow cytometry, the expression of apoptosis-related proteins displayed significantly increased expression levels of caspases 3 and 7 after treatment with the **RuL** complex (Figure 6). Furthermore, the protein level of caspase 3 was significantly increased after treatment compared with untreated cells. The findings indicated that the ruthenium complex induced HCC cell apoptosis primarily through the activation of caspase-mediated apoptosis due to the proteolysis of the important cellular proteins. These results are consistent with previously reported studies that their investigated ruthenium complexes also induced the activation of the caspase-mediated apoptosis pathway [38,41].

#### 3.3.4. Effect of the RuL Complex on H2AX Expressions

It is well known that apoptosis activation and subsequent DNA cleavage represent the main cytotoxic mode of action of metal-based antiproliferative drugs [42]. The current investigation showed that HepG2 cell exposure to the **RuL** complex led to an increased proportion of apoptotic and necrotic cells. The complex resulted in the overexpression of the H2AX expression level (Figure 7), which in turn confirmed the presence of DNA damage in HepG2. The present results corroborate previous studies that reported the induced apoptosis by ruthenium complexes [43,44].

#### 3.3.5. Effect of RuL Treatment on VEGF in HepG2 Cells

Vascular endothelial growth factor A (VEGF-A) is the main proangiogenic factor in development, wound healing, and pathogenic conditions, such as carcinogenesis [45]. It is highly expressed in cancer tissue and correlates with its more aggressive features. New blood vessel formation (angiogenesis) is a fundamental event in the process of tumor growth and metastatic dissemination [46]. In addition, the prosurvival activity of VEGF requires the phosphatidylinositol 3-kinase (PI 3-kinase)/Akt signal transduction pathway. Results of the current study manifested a significant reduction in the gene expression and protein level of VEGF-A in the **RuL**-treated cells as compared with the untreated ones (Figure 8A,B). These findings supported the antiangiogenic activity of the **RuL** complex. In vitro studies of many ruthenium complexes have been previously identified as an inhibitor of angiogenesis [47,48].

#### 3.3.6. Effect of the RuL Complex on AKT and mTOR of HepG2 Cells

The mammalian target of rapamycin (mTOR) is a highly conserved serine/threonine kinase that is frequently found in cancer and is regulated by several upstream regulators, including PI3/Akt [49]. The activation of the PI3K/Akt/mTOR signaling pathway has been linked to the promotion of VEGF-A-induced proliferation, which in turn facilitates angiogenesis [50]. VEGF and PI3K/AKT/mTOR were two main pathways implicated in the carcinogenic process in HCC [51]. Besides VEGF-A, the mammalian target of the rapamycin (mTOR) signaling pathway was also important in the regulation of cell proliferation, migration, survival, and angiogenesis, making it a promising target in HCC. mTOR plays an important role in regulating apoptosis and autophagy [52]. Results of this study revealed that **RuL** inhibited angiogenesis by targeting the VEGF-A-mediated Akt/mTOR signaling pathway. **RuL** also resulted in the downregulation of the expression of Akt and mTOR as well as the protein of mTOR in HepG2 cells compared with untreated cells (Figure 9). The availability of VEGF, the critical modulator of tumor angiogenesis, was reduced as a result of such an influence on the expression level of these proteins. These findings showed that **RuL** disrupted several signaling pathways, resulting in changes in the activation and/or production of proteins involved in invasion and metastasis.

#### 3.3.7. Effect of the RuL Complex on SND1 of HepG2 Cells

The protein staphylococcal nuclease domain-containing protein 1 (SND1) is involved in several cellular biological processes, including gene transcription, pre-mRNA splicing, cell cycle, repair DNA damage, proliferation, programmed cell death, adipogenesis, and carcinogenesis [53,54,55,56]. Overexpression of SND1 promotes tumorigenesis in many types of cancer, including HCC [57,58,59]. Furthermore, it was found that the SND1 protein activated NF-kB, increased angiogenesis, and promoted human HCC cell migration and invasion [60], as well as played a role in DNA damage [56] and the antiapoptotic pathway in HepG2 [61]. SND1 contributes to the hallmarks of cancer through several mechanisms, resulting in tumor aggressiveness and chemoresistance to various chemotherapeutic drugs. **RuL**-treated HepG2 cells reduced the expression of the mRNA gene and protein level of the SND1 compared with untreated HepG2 cells (Figure 10). It was suggested that SND1 functions as an oncogene in HCC and contributes to cancer by a variety of mechanisms, implying that targeting SND1 could be a viable treatment option for HCC treatment [62].

#### 3.3.8. Effect of the RuL Complex on NF-kB in HepG2 Cells

Nuclear factor kappa B (NF-kB) is a key signaling pathway that regulates hundreds of genes and is involved in a variety of events, including cell proliferation, differentiation, development, and death [52,63]. Our study conveyed significant inhibition in the mRNA expression and protein level of NF-kB after treatment with the **RuL** complex, as shown in Figure 11. NF-kB activation has been linked to many essential characteristics of carcinogenesis, including apoptosis inhibition, cancer initiation, tumor cell proliferation, and tumor development [64]. Inhibiting NF-kB-sensitized cancer cells to treatment is considered one of the important targets for the development of chemotherapeutic agents.

### 3.4. Molecular Docking of the Ruthenium (RuL) Complex

Molecular docking is an excellent approach to understand the interaction between the synthesized compounds and biological target. Analysis of the docking data is useful in predicting the conformational changes associated with the amino acid residues at the binding position to accommodate the docked hydrophobic inhibitors. The ruthenium complex (**RuL**) was subjected to molecular docking studies using the MOE, version 2014.09, to understand the complex–targets interactions (binding poses) and to explore the potential binding mode and energy. The docked complex conformations were rated according to the binding affinity, hydrogen bonding, and hydrophobic interactions between the complex and macromolecule. The macromolecule targets were: a B-DNA (PDB: 1BNA) and caspase 3 protein (PDB: 3KJF). These two macromolecular receptors have been chosen according to the experimental finding, which declared that the complex has the best effect on the action of DNA and caspase 3 activity (vide supra). The docking studies determine the way by which the docked complex fundamentally fits in the macromolecule minor groove and comprises hydrophobic, ionic, and hydrogen bonding interactions with the target bases. Figure 12 illustrates 2D and 3D molecular docking interaction representations of the ruthenium complex with the studied targets. The values of final score values (S) of the interactions of the ruthenium complex with the B-DNA and caspase 3 protein targets, which are related to the binding affinity, were −3.25 and −3.91 kcal/mol, respectively. Thus, the **RuL** complex showed good binding scores with high negative values, which indicated high binding affinity between the receptors and the complex molecule. In addition, these values illustrate that the complex has high efficiency as a bioactive compound. In the case of the 1BNA receptor, the binding interactions with the DNA came from either hydrophobic interactions between the amino acid residues (DG 22 and DA 5) or hydrogen bonding between the DA 6 region and both the coordinated water and nitrogen atoms of the coordinated Schiff base ligand (Figure 12). Molecular docking of previously reported ruthenium complexes in its different oxidation states (0, II, and III) showed variable binding affinities to the DNA. The score values varied between −9.0 and −5.0 according to the oxidation state of metal species [11,65,66,67,68]. Although the current ruthenium complex displayed a lesser S value, it was characterized by the presence of three H-bonding. Such interaction could be responsible for the higher activity towards DNA as indicated from the experimental data. It is worth mentioning that the most binding interaction of the ruthenium complexes was via the DA residues of the nucleic acid target. In the case of the caspase 3 protein (3KJF), the docked ruthenium molecule also showed a good binding score (S = −3.91) as it formed hydrogen bonding with an Asp 211 part and hydrophobic interactions with Lys 210 and Gln 217 moieties (Figure 12). To our knowledge, there were no previous molecular docking data reported for the binding between ruthenium complexes and the caspase 3 protein (3KJF). Thus, it is obvious that these bioactive compounds were able to interact effectively with the available binding sites of the studied targets. In addition, the molecular docking studies supported the experimental findings. The increased value of total apoptosis of treated HepG2 cells with the **RuL** complex relative to untreated ones could be due to the strong interaction of the complex with the DNA. Furthermore, the observed elevation in the activity of treated caspase 3 with the ruthenium complex can be regarded due the possible binding interaction. The ability of the **RuL** complex to activate caspase 3 could be presumably demonstrated by the cell permeability concept and/or Tweedy’s chelation theory [69,70,71]. The cell permeability concept declared that the polarity of the chelated metal ions is significantly reduced and improves the lipophilicity property, and consequently, the formed **RuL**-caspase 3 species would increase the caspase 3 activity.

## 4. Conclusions

A novel ruthenium–Schiff base complex was found to be a potent inhibitor of progression in HepG2 cells via the induction of apoptosis and necrosis. In vitro studies indicated that the complex induced cell cycle arrest and apoptosis through damage of DNA and the inhibition of the cell growth in the S phase pathway. Further, the complex increased the apoptosis-related gene expression and possessed antiangiogenic activity by inhibiting VEGF, AKT/mTOR pathway, SND1, and NF-kB gene expression. Molecular docking studies of the complex supported the increased total apoptosis of the treated HepG2 cells via strong interaction with DNA. The possible binding interaction of the Ru complex with caspase 3 could also be responsible for enhancing the activity of caspase 3 in treated HepG2 cells.

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
