# Peer review of "Molecular Design, Spectroscopic, DFT, Pharmacological, and Molecular Docking Studies of Novel Ruthenium(III)–Schiff Base Complex: An Inhibitor of Progression in HepG2 Cells"

_ijerph, 2022, doi:10.3390/ijerph192013624_

Round 1
Reviewer 1 Report
The work is done nicely but only docking part seems to be extensively studied so some work has to be done on other part too, for example,
In Figure 2 why HOMO and LUMO are identical?
In basis set why to used different basis set for ligand and for complex?
In Figure 3S Scale not clear and IR spectra seems to be not discussed in details or compared to calculated ones (I suggest doing a frequency calculation). And why both IR spectra of ligand and of complex are so similar?
IN NMR spectra what is the assignment for the band at 3.25
My main concern about this research is the chemical structure of the complex
Which is as follows
1- Water and chlorine in complex can be confirmed by conductance which has not been done!
2- Inside water can be confirmed by thermal analysis which has not been done!
None of 1 or 2 can be proved by theoretical unless you do calculation for all possibilities and show the relative stability which has not been done
So structure is not proved to be true.
Author Response
Reviewer 1:
The work is done nicely but only docking part seems to be extensively studied so some work has to be done on other part too, for example,
In Figure 2 why HOMO and LUMO are identical?
- Fixed
In basis set why to used different basis set for ligand and for complex?
- On running Gaussian calculations for non-transition elements, the basis set 6-31G (d,p) is used, while for compounds having transition metals such as Ru, the basis set LANL2DZ is used.
In Figure 3S Scale not clear and IR spectra seems to be not discussed in details or compared to calculated ones (I suggest doing a frequency calculation). And why both IR spectra of ligand and of complex are so similar?
- The figure 3S for IR spectra is enhanced and the scale are now clear. It is obvious that the two spectra are almost similar in the region 1700-1400 cm-1 with the appropriate shift due to complex formation, while they are different in the lower wavenumbers as fingerprint to every compound. However, declare of comment is added in discussion.
IN NMR spectra what is the assignment for the band at 3.25
- This signal is due to the DMSO solvent.
My main concern about this research is the chemical structure of the complex
Which is as follows
- Water and chlorine in complex can be confirmed by conductance which has not been done!
- The molar conductance values for the complex are measured and the complex is found to be non-electrolyte. Data is given in text. The elemental analysis content of Cl in complex is carried out and data is inserted in text.
2- Inside water can be confirmed by thermal analysis which has not been done!
None of 1 or 2 can be proved by theoretical unless you do calculation for all possibilities and show the relative stability which has not been done
So structure is not proved to be true.
- The presence of a water molecule in the complex is indicated from its experimental mass spectrum as it gives a clear signal at m/z = 500 due to the ion (P-H2O)+. Declared in text.
Reviewer 2 Report
Authors have evaluated the ruthenium (III)-Schiff base complex for its pharmacological and docking studies. The manuscript is interesting for publication after some revisions suggested below:
Abstract: Quantitative information is missing in the abstract. Authors are advised to include some quantitative information in order to enhance the readability of the manuscript.
Introduction: I suggest may be revise introduction should be still more information about molecular docking studies in cancer.
Formatting: Authors are strongly advised to follow the journal guidelines and style of the journal. Several formatting issues must be resolved before its publication.
Abbreviations: Authors are suggested to define each abbreviation in the abstract, text, table and figure in its first appearance. Continue using abbreviated form after its definition.
The discussion should go into further depth, discussing your current findings and concluding with future directions, as well as the advantages of this work in cancer patients' medication treatments.
Space symbol is missing throughout the manuscript.
The quality of the Figures 3 and 4 should be improved.
The language should be carefully modified.
References: The references are not formatted according to journal style.
Author Response
Reviewer 2:
Authors have evaluated the ruthenium (III)-Schiff base complex for its pharmacological and docking studies. The manuscript is interesting for publication after some revisions suggested below:
Abstract: Quantitative information is missing in the abstract. Authors are advised to include some quantitative information in order to enhance the readability of the manuscript.
- OK, some quantitative data are given in abstract.
Introduction: I suggest may be revise introduction should be still more information about molecular docking studies in cancer.
- More information about the use of molecular docking studies in cancer drugs are added in Introduction section.
Formatting: Authors are strongly advised to follow the journal guidelines and style of the journal. Several formatting issues must be resolved before its publication.
- Done according to Journal.
Abbreviations: Authors are suggested to define each abbreviation in the abstract, text, table and figure in its first appearance. Continue using abbreviated form after its definition.
- Abbreviations are defined in its first appearance.
The discussion should go into further depth, discussing your current findings and concluding with future directions, as well as the advantages of this work in cancer patients' medication treatments.
- Declared in text.
Space symbol is missing throughout the manuscript.
The quality of the Figures 3 and 4 should be improved.
- Better resolved figures (3 and 4) are inserted.
The language should be carefully modified.
- Paper was revised for any possible errors or mistakes.
References: The references are not formatted according to journal style.
- References are made according to journal style.
Round 2
Reviewer 1 Report
Thanks for the replies please note thank accuracy and clarity of paper is your own responsability.